# Trends in Systemic Inflammatory Reaction (SIR) during Paclitaxel and Carboplatin Chemotherapy in Women Suffering from Epithelial Ovarian Cancer

**DOI:** 10.3390/cancers15143607

**Published:** 2023-07-13

**Authors:** Michal Mleko, Elzbieta Pluta, Kazimierz Pitynski, Maciej Bodzek, Andrzej Kałamacki, Dorota Kiprian, Tomasz Banas

**Affiliations:** 1Department of Gynaecology and Gynaecological Oncology, Faculty of Medicine, Jagiellonian University Medical College, 31-501 Krakow, Poland; 2Department of Radiotherapy, Maria Sklodowska-Curie National Research Institute of Oncology, Kraków Branch, 31-115 Krakow, Poland; 3Department of Gynaecology and Obstetrics, Faculty of Medicine and Health Sciences, Andrzej Frycz Modrzewski Krakow University, 30-701 Krakow, Poland; 4Department of Gynaecology Oncology, Maria Sklodowska-Curie National Research Institute of Oncology, Kraków Branch, 31-115 Krakow, Poland; 5Radiotherapy Department I Maria Sklodowska-Curie National Research Institute of Oncology, Warszawa Brand, 02-781 Warsaw, Poland

**Keywords:** chemotherapy, systemic inflammatory reaction, epithelial ovarian cancer, platinum, taxanes

## Abstract

**Simple Summary:**

Epithelial ovarian cancer remains the most fatal gynaecological malignancy, and cytoreductive surgery followed by adjuvant taxane-platinum-based chemotherapy remains the core therapy for women suffering from this cancer. The systemic inflammatory response plays a dual role in the pathophysiology of neoplastic diseases, activating the cytotoxic immune response against cancer cells and contributing to the progression of the disease via inflammatory mediators, including cytokines and growth factors. Increased systemic inflammatory markers at the time of cancer diagnosis are associated with advanced stage and tumour grade and predict poor survival; however, only a few studies have investigated changes in inflammatory markers during anti-cancer treatment in the context of clinical and pathological cancer features as well as applied therapies. In this study, we aimed to investigate the trends in changes in inflammatory markers in women with ovarian cancer receiving standard first-line adjuvant chemotherapy and identify the potential factors influencing these changes.

**Abstract:**

Background: Epithelial ovarian cancer (EOC) is the most fatal gynaecological malignancy treated with cytoreductive surgery followed by adjuvant taxane-platinum-based chemotherapy. It has been shown that the pretreatment systemic inflammatory reaction (SIR) in women with OC can be evaluated using the neutrophil-to-lymphocyte ratio (NLR), lymphocyte-to-monocyte ratio (MLR), platelet-to-lymphocyte ratio (PLR) and systemic inflammatory index (SII), depending on the stage of disease, and has prognostic value for overall survival. The aim of this study was to evaluate the changes in NLR, LMR, PLR and SII during chemotherapy. Methods: A total of 107 women with EOC (23 with type I and 84 with type II tumours) were included in a retrospective single-centre analysis. The Kologomorov−Smirnoff, Kruskal-Wallis or Friedman analysis of variance tests were used for data analysis, and a *p* value of 0.05 was considered statistically significant. Results: A significant decrease in NLR, PLR and SII but not LMR was observed during adjuvant treatment. Pretreatment NLR, PLR and SII were dependent on disease stage and tumour grade; however, this association was lost during therapy. Additionally, strong and positive mutual correlations between NLR, LMR, PLR and SII were sustained during the whole course of chemotherapy. Conclusions: During first-line adjuvant chemotherapy in women with EOC, a decrease in SIR is confirmed.

## 1. Introduction

Epithelial ovarian cancer (EOC) is the most fatal gynaecological malignancy [1]. Based on pathological and molecular features, two types of EOC are distinguished. Type I, which includes low-grade serous, mucinous, endometrioid and clear-cell carcinomas, accounts for 25% of all EOCs, is less aggressive, tends to grow more slowly and has a more favourable prognosis [2,3,4]. On the other hand, high-grade type II EOC, which accounts for 75% of all EOCs, comprises mostly serous tumours characterised by rapid growth and aggressive spread [2,3,4]. Due to nonpathognomic symptoms and a lack of screening, EOCs are often diagnosed in their advanced stages, which results in a poor prognosis [2]. Over 60% of women with EOC at the time of diagnosis suffer from stage III or IV according to the International Federation of Gynaecology and Obstetrics (FIGO)’s classification, which means that the tumour is not only limited to the ovaries (stage I) or to the uterus and pelvis (stage II) but there is also cancerous spread that involves the upper peritoneal cavity and/or regional lymph nodes (stage III) or distant metastases (stage IV) [5]. Therefore, cytoreductive surgery is the core therapy for ovarian cancer, followed by taxane- and platinum-based first-line chemotherapy [5]. In developed female populations, ovarian cancer is the third most common malignancy of the reproductive tract but has the highest mortality rate [6,7,8].

Chronic inflammation may contribute to the development of many cancers due to its dual role in the pathophysiology of neoplastic diseases. On the one hand, it offers protection, but on the other hand, it contributes to the progression of the cancer via inflammatory mediators, including cytokines and growth factors [9,10,11,12]. The severity of a systemic inflammatory reaction (SIR) can be evaluated using various biochemical markers, including levels of C-reactive protein (CRP), albumins and blood counts. The ratios of particular types of white blood cells, such as the lymphocyte-to-neutrophil ratio (NLR), the platelet-to-lymphocyte ratio (PLR) and the lymphocyte-to-monocyte ratio (LMR), as well as the systemic immune-inflammation index (SII) and CRP-to-albumin ratio (CAR), are important predictive markers in various malignancies, including ovarian cancer [9,10,11,12].

In this study, we aimed to evaluate changes in SIR during first-line postoperative chemotherapy based on platinum and taxanes in women suffering from ovarian cancer.

## 2. Materials and Methods

Identification of patients: We initially identified 156 patients with ovarian cancer who were treated in our tertiary gynaecologic oncology unit. The inclusion criteria for further analysis were as follows: (1) epithelial ovarian cancer; (2) age 18+ years; (3) primary debulking surgery with complete (R0) resection; and (4) full histopathological report confirming ovarian cancer diagnosis and full staging. Patients with (1) incomplete medical records, (2) interval debulking surgery or (3) nonepithelioid ovarian malignancy, as well as those (4) diagnosed with a malignancy in addition to ovarian cancer, (5) requiring blood transfusion during chemotherapy, (6) experiencing cytotoxicity resulting in chemotherapeutic regimen modification, (7) developing NADIR or (8) suffering from autoimmune diseases, were excluded from this study. From the total number of 156 identified patients with ovarian malignancy, 107 women (68.59%) were eligible for this study after applying the inclusion and exclusion criteria.

Therapeutic procedures: All included patients had undergone primary radical cytoreductive surgery to achieve R0 resection, followed by standard first-line adjuvant chemotherapy. During adjuvant treatment, every patient received intravenous paclitaxel (175 mg/m^2^) with the addition of carboplatin (AUC = 5) every 3 weeks. Standard premedication with H1 blockers and corticosteroids was applied on a routine basis.

Evaluation of SIR: Peripheral blood samples were drawn from the ulnar vein (3.0 ML) a day prior to chemotherapy onset (sample 1) and then before each treatment cycle (samples 2–6) and placed in tubes with ethylenediaminetetraacetic acid (EDTA). Haemoglobulin (Hb) and haematocrit (Hct) levels, erythrocyte, total lymphocyte, monocyte, eosinophil and basophil counts, as well as total platelet counts, were evaluated. The neutrophil-to-lymphocyte ratio (NLR) was defined as the neutrophil and lymphocyte quotient; the lymphocyte-to-monocyte ratio (LMR) and platelet-to-lymphocyte ratio (PLR) were calculated by dividing lymphocyte values by monocyte values and platelet values by lymphocyte values; and the systemic inflammatory index (SII) was defined as neutrophil count multiplied by platelet count and divided by lymphocyte count.

Statistical analysis: Using the Kolmogorov−Smirnoff test, the distribution of the continuous variables analysed was checked. Data presenting a normal distribution were presented as medians and standard deviation (±SD), while variables with a distribution different from normal, as well as increment data, were shown as the means and interquartile range (IQR). Categorised variables are shown as the number of cases (n) and a percentage (%) and were compared using the chi-square test. Kruskal-Wallis or Friedman analysis of variance (ANOVA) was used to evaluate differences between more than two study groups, with a post hoc analysis if needed. Spearman test was used to evaluate correlations between independent variables, while gamma test was applied to related ones. Additionally, the Newman test was employed for trend analysis of SIR markers and Ca125 levels. A *p* value of 0.05 was considered statistically significant, and all the calculations were performed using STATISTICA data analysis software (TIBCO Software Inc. 2017, version 13.0, Palo Alto, CA, USA).

## 3. Results

### 3.1. Patient Characteristics

The mean age of the participants was 64,52 years (±8.72), and the vast majority were postmenopausal (n = 69; 64.49%). Type I EOC was confirmed in 23 (21.50%) women, while 84 (78.50%) patients suffered from Type 2 cancer. Table 1 presents the detailed characteristics of the study population. The median interval between surgery and chemotherapy was 27 days (IQR: 3.5).

### 3.2. Ca125 Trends during Therapy

During chemotherapy, a significant decreasing trend in the median Ca125 values was confirmed in both low- and high-grade patients, while in the second group it was more dynamic (Table 2; Figure 1). The median Ca125-1, Ca125-2, Ca125-3 and Ca125-4 values in the women with high-grade EOC were significantly higher compared to the median Ca125 values in the low-grade patients (Table 2). No significant differences were observed between low- and high-grade EOC patients for Ca125-5 and Ca125-6 levels (Table 2). In the women with low-grade EOC, median Ca125-1 and Ca125-2 levels were found to be significantly higher compared only to the median Ca-125-6 value (*p* = 0.019 and *p* = 0.037, respectively). In the high-grade EOC patients, the median Ca125-1 level was also significantly higher compared to the Ca125-3, Ca125-4, Ca125-5 and Ca125-6 values (*p* = 0.001, *p* < 0.001, *p* < 0.001 and *p* < 0.001, respectively). Similarly, the median Ca125-2 values in the women with high-grade EOC were significantly higher compared to the median Ca125-4, Ca125-5 and Ca125-6 values (*p* < 0.001, *p* < 0.001 and *p* < 0.001, respectively). Additionally, Ca125-3 levels in this group were significantly higher compared to Ca125-5 and Ca125-6 (*p* = 0.003 and *p* < 0.001, respectively), while the Ca125-4 level was significantly higher only compared to the Ca125-6 value (*p* = 0.048) (Table 2).

The median Ca125-1 level in the women with FIGO III EOC was significantly higher than the median Ca125-1 levels in the patients with EFIGO I (*p* = 0.040) and FIGO II (*p* = 0.004) EOC (Table 2). In the FIGO II women, significant differences were observed between the median Ca125-1 and Ca125-5 (*p* = 0.003) values and the Ca125-1 and Ca125-6 values (*p* < 0.001) (Table 2).

In the EOC FIGO stage III women, the median Ca125-6 level was significantly higher than the median Ca125-1 (*p* < 0.001), Ca125-2 (*p* < 0.001) and Ca125-3 (*p* = 0.001) levels (Table 2). Similarly, the median Ca125-5 level in this group was significantly higher than the median Ca125-1 (*p* < 0.001) and Ca125-2 (*p* = 0.001) levels (Table 2). Additionally, significant differences were observed between the median Ca125-4 and Ca125-1 (*p* < 0.001) and Ca125-2 (*p* = 0.011) in the women with EOC stage III (Table 2).

### 3.3. Neutrophil-to-Lymphocyte Ratio Trends during Therapy

In the EOC patients, we observed a decreased median NRL during first-line chemotherapy in both the low- and high-grade groups (Figure 1). Significant differences between low- and high-grade EOC patients were observed when comparing the medians of NLR-1, NLR-2 and NLR-3 (Figure 2).

Subsequent analyses revealed that the median NRL-1 level in the FIGO III patients was significantly higher than the median NRL-1 level in the FIGO I and FIGO II women (Figure 3). Similarly, the highest median NRL-2, NLR-3, NLR-4 and NLR-5 levels was confirmed in FIGO III compared to FIGO I and FIGO II women, while there were no significant differences in the median NRL-6 levels between the FIGO I and FIGO II patients (Figure 3). The only significant association was proven between the NRL-1 and Ca125-1 values (R = 0.227; *p* = 0.003) in the high-grade EOC group, while in the low-grade cancer group, no association between NLRs and Ca125 levels was observed. Additionally, moderate, positive and significant correlations between the NRL-1 and NRL-6 values were proven (Appendix A).

### 3.4. Lymphocyte-to-Monocyte Ratio Trends during Therapy

No significant differences in the median LMR levels were observed during first-line chemotherapy, neither in low- nor high-grade EOC women (Figure 1).

In contrast to NLR, we did not observe any associations between tumour grade or FIGO stage and LMR (Figure 4 and Figure 5). Similarly, there were no associations between the LMR and Ca125 levels, neither in low- nor high-grade cancer cases. However, we observed significant positive correlations between all the LMR values (Appendix A).

### 3.5. Platelet-to-Lymphocyte Ratio Trends during Therapy

The highest median PLR was confirmed before chemotherapy onset (PLR-1) and after the first drug administration (PLR-2) in high-grade EOC patients (Figure 6), with an observed significant decrease during treatment in both the low- and high-grade groups that was more dynamic in the latter (Figure 1).

The highest median PLR values were observed in high-grade tumours, with significant differences observed only in the PRL-1 and PLR-2 medians (Figure 6). In the FIGO stage III women with EOC during first-line chemotherapy, the highest median values of PLR-1 to PRL-6 were confirmed (Figure 7). However, the only significant differences were proven in the median PRL-1 values between FIGO stage III and FIGO stages I and II (Figure 7). A positive significant correlation was proven only in the high-grade EOC women between PLR-1 and Ca125-1 (R = 0.404; *p* < 0.001) and between PLR-2 and Ca125-2 (R = 0.288; *p* = 0.047), while no associations between PLR and Ca125 laboratory determinations were observed in low-grade EOCs. In contrast, we observed strong, positive and significant correlations between all the PLR values (Appendix A).

### 3.6. Systemic Inflammatory Index Trends during Therapy

In the women with EOC treated with first-line chemotherapy after primary radical surgery, a gradual decrease in the median SII values was observed in both the low- and high-grade patients (Figure 1). Significant differences between the median SII values according to tumour grading were observed only during the first two measurements (SII-1 to SII-2), while there were no significant differences between the median SII values in the second part of chemotherapy (Figure 8).

Additionally, there were positive and significant correlations between all the SII values (Appendix A). The highest median SII values were observed in the FIGO stage III women throughout the whole chemotherapy treatment, and the differences were significant compared to the FIGO stage I and FIGO stage II women (Figure 9). The only significant and positive association was observed between SII-1 and Ca125-1 in both the low- and high-grade patients (R = 0.441; *p* = 0.039, and R = 0.389; *p* < 0.001, respectively), while no other mutual correlations were substantial.

## 4. Discussion

Ca-125 is a common biomarker in patients with ovarian cancer and is of well-established significance in diagnosis and prognosis. Ca-125 levels increase in advanced-stage disease because of an increase in tumour burden [13,14,15,16,17]. During chemotherapy, Ca-125 levels decrease. An earlier lowering of the level of CA-125 was associated with better survival, while the normalisation of CA-125 prior to the second cycle instead of the third cycle of chemotherapy was associated with improved OS. [16,17] Kim et al. found that CA-125 levels were best for predicting advanced-stage disease, suboptimal debulking and platinum resistance, and that PLR and NLR may be the most effective predictors of noncomplete response and PFS in patients with OC [18]. However, recent research has shown that there is no significant correlation between them.

To our knowledge, a comprehensive investigation of the trends in SIR parameters such as NLR, LMR, PLR and SII in relation to Ca125 levels during first-line adjuvant chemotherapy in low- and high-grade EOC women has not been reported previously. So far, many studies have confirmed that these are sensitive indicators correlated with local tumour advancement as well as the response to first-line chemotherapy in ovarian cancer [18,19,20,21,22]. An elevated NLR was independently associated with worse overall survival (OS) but not progression-free survival (PFS) [18]. Similarly, Huang et al. conducted a systematic review and meta-analysis of 12 observational studies involving 3854 patients with ovarian cancer. They showed that a high pretreatment NLR was significantly associated with shorter OS and PFS [16]. An elevated NLR also correlated significantly with an advanced FIGO stage, increased serum CA-125 concentration, the degree of ascites and a poorer response to chemotherapy [19]. An increased NLR was also confirmed in ovarian cancer patients with low CA-125 levels (<35 U/mL) [15]. Similar to these results, we confirmed higher median levels of NRL in FIGO III OC patients than in women with OC stages I and II. Moreover, although a gradual decrease in median NLR levels was observed during first-line chemotherapy, significantly higher median NRL levels in FIGO III patients were maintained throughout the whole treatment.

PLR has also been proposed as an indicator of prognosis in epithelial ovarian cancer in numerous studies [10,11]. An elevated PLR at OC diagnosis was associated with worse OS, and PLR levels decreased with the use of chemotherapy [23,24]. We confirmed the highest median PLR before chemotherapy onset, with a significant decrease during treatment. We found strong, positive and significant correlations between all PLR values.

SII values are correlated with poor histological differentiation of the tumour, a larger tumour size and a more advanced TNM stage [25]. Studies have also confirmed that a high SII is associated with shortened progression-free survival and increased mortality in patients with ovarian cancer [26]. The SII is a predictor of the efficacy of neoadjuvant chemotherapy in patients with gastric cancer, cervical cancer and breast cancer [27,28,29]. However, there are still few studies on the relationship between the SII and the effect of neoadjuvant chemotherapy in patients with ovarian cancer. Wang et al. showed that the complete response rate was significantly lower in women with an increased SII after chemotherapy completion compared to low-SII-value OC patients, while the risk of progression was increased [30]. Additionally, a higher SII was a risk factor for death in ovarian cancer patients after neoadjuvant chemotherapy [31].

Moreover, we observed no significant differences in the median LMR levels during OC first-line chemotherapy. There were no associations between the median LMR values during chemotherapy and the stage of the disease or tumour grade.

The novelty of our study is the comprehensive evaluation of the trends in SIR during adjuvant standard chemotherapy in EOC patients (as opposed to only point pretreatment values). The analysis was performed in pathology-proven EOC separately for low- and high-grade tumours in patients receiving homogeneous therapy, allowing for the reduction of potential unspecific bias. Additionally, we managed to determine an interrelationship between SIR and CA125 levels and investigated the effect of cancer staging and tumour grading on the values of individual SIR markers during chemotherapy.

We are also aware that our study has some limitations that must be discussed. First, we were unable to determine the influence of the changes in SIR levels on OS and PFS, which was due to the short follow-up time of less than 36 months; however, such an analysis is planned to be carried out at a relevant time. Second, the small number of low-grade EOC cases means that the results in this group should be interpreted with caution.

## 5. Conclusions

Fluctuations in SIR markers in EOC patients during first-line standard chemotherapy were confirmed, along with a significant decrease in Ca-125 values. NLR and SII showed significantly higher values in FIGO III EOC patients throughout the whole course of chemotherapy, while significant differences in the median Ca-125 and PLR values according to the EOC stage were observed only before therapy onset. Cancer grade influenced only pretreatment NLR, PLR and SII values, while during chemotherapy, the medians of these SIR markers were independent of the cancer grade. Significant, strong and positive correlations between NLR 1–6, LMR 1–6, PLR 1–6 and SII 1–6 were observed. Additionally, NLR, PLR and SII values correlated significantly with Ca-125 levels only at the beginning of chemotherapy, while no such relationship was observed during its duration. Based on the above findings, we conclude that SIR markers during first-line chemotherapy change regardless of the Ca-125 levels and thus can be considered potential independent predictive markers of first-line adjuvant chemotherapy response in OC women. However, to verify this hypothesis, further studies are needed.

## Figures and Tables

**Figure 1 cancers-15-03607-f001:**
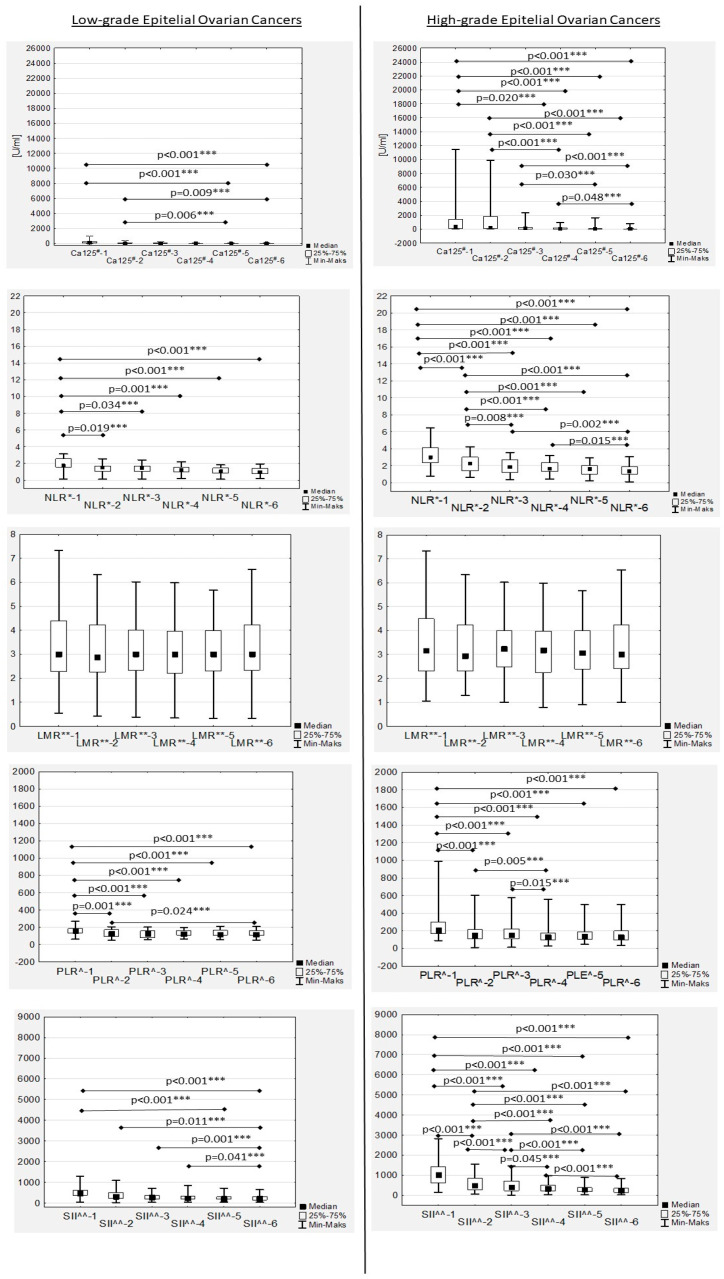
Trends in Ca125 levels and systemic inflammatory reaction (SIRS) markers during standard first-line adjuvant chemotherapy in women with ovarian cancer (EOC). Ca125^#^—Cancer antigen 125; NLR*—neutrophil-to-lymphocyte ratio; LMR**—lymphocyte-to-monocyte ratio; *** *p*-value statistically significant (<0.050); PLR^—platelet-to-lymphocyte ratio; SII^^—systemic inflammatory index.

**Figure 2 cancers-15-03607-f002:**
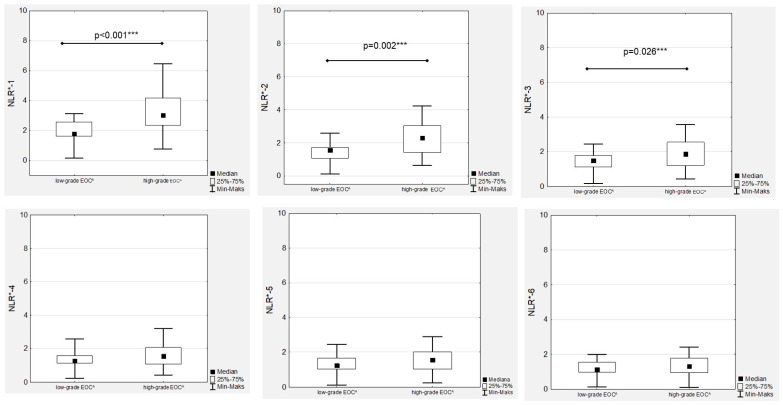
Neutrophil-to-lymphocyte ratios according to the EOC^ grade during first-line chemotherapy in ovarian cancer patients. NLR*—neutrophil-to-lymphocyte ratio; EOC^—epithelial ovarian cancer; *** *p* statistically significant.

**Figure 3 cancers-15-03607-f003:**
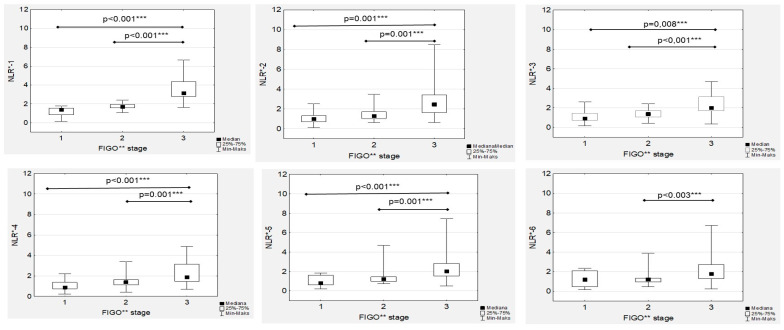
Neutrophil-to-lymphocyte ratios according to the International Federation of Gynaecology and Obstetrics (FIGO) stage during first-line chemotherapy in ovarian cancer patients. NLR*—neutrophil-to-lymphocyte ratio; FIGO**—International Federation of Gynaecology and Obstetrics; *** *p* statistically significant.

**Figure 4 cancers-15-03607-f004:**
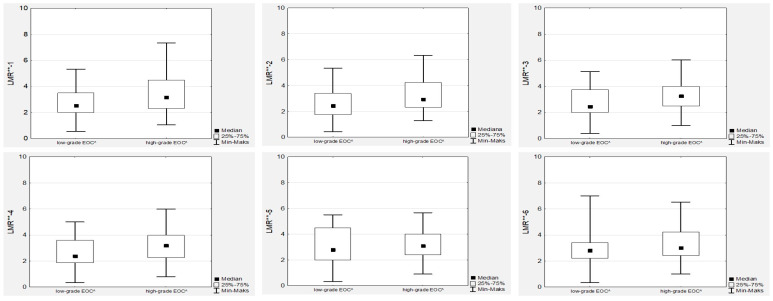
Lymphocyte-to-monocyte ratios according to the EOC^ grade during first-line chemotherapy. LMR**—lymphocyte-to-monocyte ratio; EOC^—epithelial ovarian cancer.

**Figure 5 cancers-15-03607-f005:**
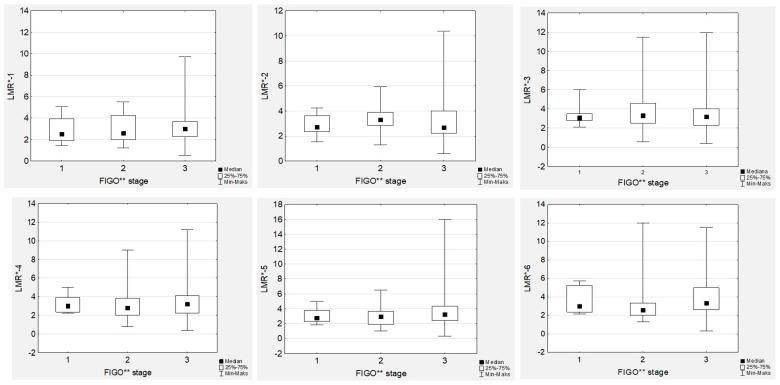
Lymphocyte-to-monocyte ratios according to the International Federation of Gynaecology and Obstetrics (FIGO) stage during first-line chemotherapy in ovarian cancer patients. LMR*—lymphocyte-to-monocyte ratio; FIGO**—International Federation of Gynaecology and Obstetrics.

**Figure 6 cancers-15-03607-f006:**
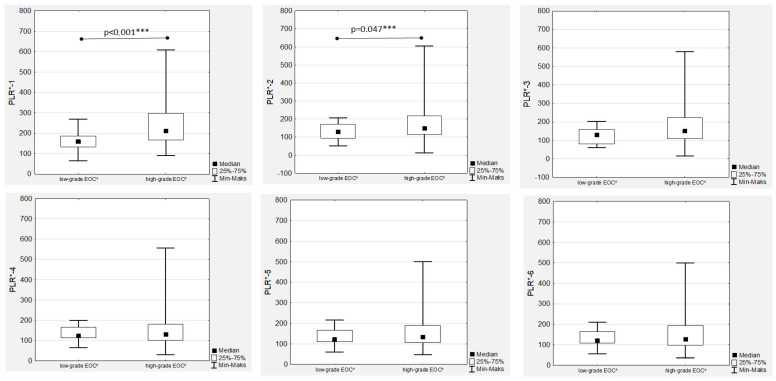
Platelet-to-lymphocyte ratios according to the EOC^ grade during first-line chemotherapy. PLR*—platelet-to-lymphocyte ratio; EOC^—epithelial ovarian cancer; *** *p* statistically significant.

**Figure 7 cancers-15-03607-f007:**
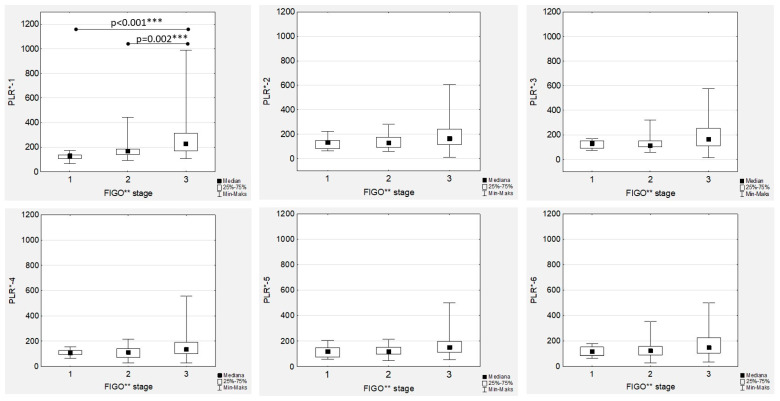
Platelet-to-lymphocyte ratios according to the International Federation of Gynaecology and Obstetrics (FIGO) stage during first-line chemotherapy in ovarian cancer patients. PLR*—platelet-to-lymphocyte ratio; FIGO**—International Federation of Gynaecology and Obstetrics; *** *p* statistically significant.

**Figure 8 cancers-15-03607-f008:**
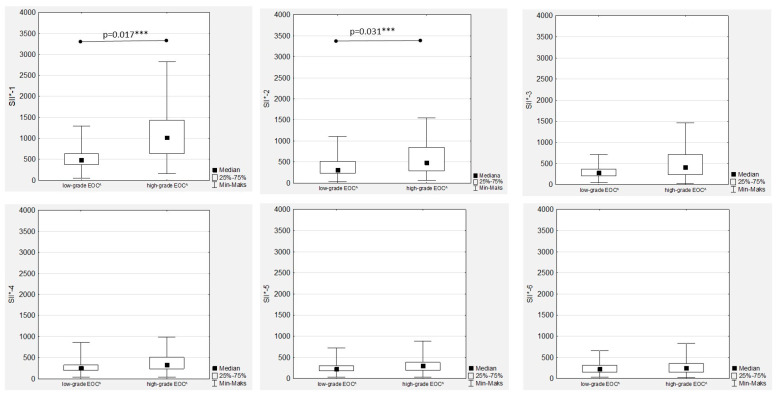
Systemic inflammatory index according to the tumour grade stage during first-line chemotherapy in ovarian cancer patients. SII*—systemic inflammatory index; EOC^—epithelial ovarian cancer; *** *p* statistically significant.

**Figure 9 cancers-15-03607-f009:**
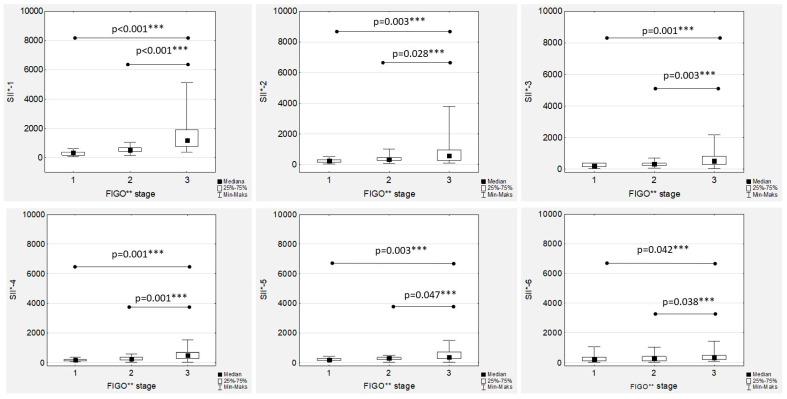
Systemic inflammatory index values according to the International Federation of Gynaecology and Obstetrics (FIGO) stage during first-line chemotherapy in ovarian cancer patients. SII*—systemic inflammatory index; FIGO**—International Federation of Gynaecology and Obstetrics. *** *p* statistically significant.

**Table 1 cancers-15-03607-t001:** Baseline characteristics of patients with epithelial ovarian cancer.

1	Mean age in years ± SD *	64.52 ± 8.72
2	Mane age at menarche in years ± SD *	12.58 ± 1.87
3	Pre-/postmenopausal (n; %)	38 (35.51%)/69 (64.49%)
4	Gravidity:	
	Nulligravida	41 (38.32%)
	Primigravida	44 (41.12%)
	Multigravida	22 (20.56%)
5	Parity:	
	Nullipara	45 (42.06%)
	Primipara	43 (40.19%)
	Multipara	19 (17.75%)
6	Histology:	
	Low-grade serous	15 (14.02%)
	Mucinous	8 (7.48%)
	Endometroid	0 (0.00%)
	High-grade serous	84 (78.50%)
7	FIGO ** stage:	
	I	8 (7.47%)
	II	28 (26.17%)
	III	71 (66.36%)
	IV	0 (0.00%)
8	Grade:	
	1	23 (21.50%)
	2	31 (28.97%)
	3	53 (49.53%)
10	Median interval between surgery and chemotherapy in days; IQR **	27 (3.50)

* SD—standard deviation; ** IQR—interquartile range.

**Table 2 cancers-15-03607-t002:** Characteristics of cancer antigen 125 (Ca-125) and SIR trends according to ovarian cancer staging.

	Epithelial Ovarian Cancer Stage according to FIGO* Classification			*p*
	I	II	III	
Median Ca125***-1 (IQR^$^)	180.35 (159.150)	197.000 (472.500)	1108.300 (1287.800)	0.001 ^
Median Ca125***-2 (IQR^$^)	72.500 (104.100)	80.650 (393.000)	105.000 (613.000)	NS**
Median Ca125***-3 (IQR^$^)	35.200 (123.300)	20.700 (62.600)	37.800 (279.200)	NS**
Median Ca125***-4 (IQR^$^)	19.400 (68.400)	17.250 (22.250)	28.200 (82.500)	NS**
Median Ca125***-5 (IQR^$^)	17.8500 (23.100)	12.250 (8.500)	20.7500 (68.450)	NS**
Median Ca125***-6 (IQR^$^)	15.350 (9.500)	12.950 (27.250)	16.200 (18.250)	NS**
	*p* < 0.001 ^	*p* = 0.002 ^	*p* < 0.001 ^	
	**Epithelial Ovarian Cancer Grade**			** *p* **
	**Low-Grade**		**High-Grade**	
Median Ca125***-1 (IQR^$^)	102.00 (187.30)		309.70 (1327.25)	0.013 ^
Median Ca125***-2 (IQR^$^)	34.00 (63.60)		182.80 (1740.70)	0.001 ^
Median Ca125***-3 (IQR^$^)	24.20 (80.00)		61.20 (150.60)	0.023 ^
Median Ca125***-4 (IQR^$^)	15.00 (22.80)		29.90 (66.50)	0.021 ^
Median Ca125***-5 (IQR^$^)	15.00 (25.50)		18.65 (44.90)	NS**
Median Ca125***-6 (IQR^$^)	15.20 (20.3)		15.70 (14.60)	NS**
	*p* < 0.001 ^		*p* < 0.001 ^	

^ *p* calculated using the Kruskal-Wallis test comparing variables with different-from-normal distribution in more than 2 groups when the post hoc range test was not needed; FIGO*—International Federation of Gynaecology and Obstetrics; NS**—not significant differences; Ca125***—Cancer antigen 125; IQR^$^—Interquartile range.

## Data Availability

The data analysed during the current study are available from the corresponding author upon request.

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
