# Peer review of "Trends in Systemic Inflammatory Reaction (SIR) during Paclitaxel and Carboplatin Chemotherapy in Women Suffering from Epithelial Ovarian Cancer"

_cancers, 2023, doi:10.3390/cancers15143607_

Round 1
Reviewer 1 Report
Interesting paper. However, these epithelial tumors need to be sub-divided into serous, endometrioid, mucinous tumors etc. In addition, it would be important to know whether any of the serous tumors were low grade and how this correlated with changes in the SIR.
I am not sure that Fig 1 correlates with the inclusion and exclusion criteria.
Author Response
25 June 2023
Dear Reviewer,
.
Thank you for reviewing our paper and your valuable comments. Please find a point by point response below. The appropriate changes in the manuscript were printed in red
Sincerely,
Tomasz Banas, MD, PhD, MPH
Interesting paper. However, these epithelial tumors need to be sub-divided into serous, endometrioid, mucinous tumors etc. In addition, it would be important to know whether any of the serous tumors were low grade and how this correlated with changes in the SIR.
@ Thank you for this important comment – in the revised version of the manuscript the epithelial ovarian tumours were divided as low-grade and high-grade cases and subsequent results were shown in the rewritten “Results” section according to your valuable remarks. All the changes were printed in red in the revised version of the manuscript.
I am not sure that Fig 1 correlates with the inclusion and exclusion criteria.
@ Thank you for this remark - the citation of figures in the text has been corrected.
Reviewer 2 Report
The manuscript “Trends in systemic inflammatory reaction (SIR) during paclitaxel and carboplatin chemotherapy in women suffering from ovarian cancer” by Michal Mleko and co-authors to investigate the trends in changes of inflammatory markers in women with ovarian cancer receiving the standard first line adjuvant chemotherapy and try to identify potential factor influencing these changes. Some concerns that mustbe taken into account before the work can be reconsidered for publication.
Comment
Do author analysis cancer antigen 125 (Ca-125) and SIR trends according to ovarian cancer type and chemoresponse?
Author Response
25 June 2023
Dear Reviewer,
.
Thank you for reviewing our paper and your valuable comments. Please find a point by point response below. The appropriate changes in the manuscript were printed in red
Sincerely,
Tomasz Banas, MD, PhD, MPH
The manuscript “Trends in systemic inflammatory reaction (SIR) during paclitaxel and carboplatin chemotherapy in women suffering from ovarian cancer” by Michal Mleko and co-authors to investigate the trends in changes of inflammatory markers in women with ovarian cancer receiving the standard first line adjuvant chemotherapy and try to identify potential factor influencing these changes. Some concerns that mustbe taken into account before the work can be reconsidered for publication.
@ Thank you for the positive feedback on our study.
Do author analysis cancer antigen 125 (Ca-125) and SIR trends according to ovarian cancer type and chemoresponse.
@ Thank you for this very important remark – In the revised version of the manuscript, following your comments, we divided the analysed population into two groups of patients ie. women with low-grade EOC (n=23) and high-grade EOC (n=83). Subsequently, appropriate statistical analyses were performed and we performed our results separately for low-grade EOC and high-grade EOC group. Unfortunately due to lack relevant information in our databased that was extracted and anonymized for this research we were unable to analyse Ca125 and SIR trends according to chemorespone however this a very interesting issue and we believe to investigate it shortly.
Reviewer 3 Report
Major objection
The authors studied the development of inflammatory markers in women with epithelial ovarian cancer who received standard adjuvant chemotherapy and sought to identify possible factors influencing these changes.
The dualistic model of ovarian carcinogenesis groups the major histopathologic subtypes into type I and type II based on clinical, genetic, and developmental components. In terms of diagnosis, approximately 30% of diagnosed ovarian cancers are type I and 70% are type II. Type I tumors are usually confined to the ovaries (stage I) and have a favorable prognosis; they account for 10% of ovarian cancer deaths. In contrast, the more aggressive II-type tumors are diagnosed at advanced stages (III, IV) where a cure is unlikely. Serum CA125 levels are elevated in 50% of early-stage tumors, which are usually type I ovarian cancers, and in 92% of advanced-stage tumors, which are usually type II ovarian cancers.
High-grade serous ovarian carcinomas generally have the strongest immune response compared with other epithelial ovarian carcinomas. The immune context and density of tumor-infiltrating T lymphocytes (TILs) vary considerably among different epithelial ovarian carcinomas, being highest in high-grade serous carcinomas, intermediate in endometrioid ovarian carcinomas, and lowest in low-grade serous ovarian carcinomas, mucinous ovarian carcinomas, and clear cell ovarian carcinomas.
The authors did not mention the histological types in their analyzed patient group... so it is not clear whether these results are from an approximately equal number of different types of epithelial ovarian cancer or whether most patients were diagnosed with high-grade serous carcinoma. In my opinion, it is wrong to compare the inflammatory response in different diseases, because epithelial ovarian carcinomas are a group of different diseases. At least a table with the clinical characteristics of the patients (proportion of high-grade serous carcinoma and other epithelial ovarian carcinomas) would be desirable.
Minor objections
1. In the introduction there is the term “developed female population“...probably means the female population in developed countries.
2. Figure 5,7,9 ...in the description, there is a term “tumor grade stage“...tumor grade is a grade of differentiation of tumor cells, and stage explains the spreading of a tumor, so we cant use both terms together.
3. Row 32..there is a tip-feller “ malignancytreated“
Author Response
25 June 2023
Dear Reviewer,
.
Thank you for reviewing our paper and your valuable comments. Please find a point by point response below. The appropriate changes in the manuscript were printed in red.
Sincerely,
Tomasz Banas, MD, PhD, MPH
The authors studied the development of inflammatory markers in women with epithelial ovarian cancer who received standard adjuvant chemotherapy and sought to identify possible factors influencing these changes.
@ We completely agree with the above.
The dualistic model of ovarian carcinogenesis groups the major histopathologic subtypes into type I and type II based on clinical, genetic, and developmental components. In terms of diagnosis, approximately 30% of diagnosed ovarian cancers are type I and 70% are type II. Type I tumors are usually confined to the ovaries (stage I) and have a favorable prognosis; they account for 10% of ovarian cancer deaths. In contrast, the more aggressive II-type tumors are diagnosed at advanced stages (III, IV) where a cure is unlikely. Serum CA125 levels are elevated in 50% of early-stage tumors, which are usually type I ovarian cancers, and in 92% of advanced-stage tumors, which are usually type II ovarian cancers.
@ Thank you very much for this valuable remarks – following your comment the data were reanalysed separately in low-grade epithelial ovarian cancers and in high-grade epithelial ovarian cancers and the “Results” section was rewritten to present relevant information – all chages are printed in red in the revised version of the manuscript
The authors did not mention the histological types in their analyzed patient group... so it is not clear whether these results are from an approximately equal number of different types of epithelial ovarian cancer or whether most patients were diagnosed with high-grade serous carcinoma. In my opinion, it is wrong to compare the inflammatory response in different diseases, because epithelial ovarian carcinomas are a group of different diseases. At least a table with the clinical characteristics of the patients (proportion of high-grade serous carcinoma and other epithelial ovarian carcinomas) would be desirable
@ Thank you for this valuable remark – relevant table (Table 1) was added in the revised version of the manuscript
In the introduction there is the term “developed female population“...probably means the female population in developed countries.
@ Thank you for this interesting comment – in the revised version of the manuscript this sentence was rewritten.
Figure 5,7,9 ...in the description, there is a term “tumor grade stage“...tumor grade is a grade of differentiation of tumor cells, and stage explains the spreading of a tumor, so we cant use both terms together.
@ Thank you for this interesting comment – table footprints were corrected.
Row 32..there is a tip-feller “ malignancytreated“
@ Thank you for this remark – this tip-feller was corrected to ”malignancy treated”.
Round 2
Reviewer 2 Report
The revised manuscript “Trends in systemic inflammatory reaction (SIR) during paclitaxel and carboplatin chemotherapy in women suffering from ovarian cancer” have adequately addressed my previous concerns and the paper is now acceptable for publication.
Reviewer 3 Report
The improved manuscript is acceptable for publishing.